# Metformin Attenuates Neutrophil Recruitment through the H3K18 Lactylation/Reactive Oxygen Species Pathway in Zebrafish

**DOI:** 10.3390/antiox13020176

**Published:** 2024-01-30

**Authors:** Ren Zhou, Rui-Chen Ding, Qian Yu, Cheng-Zeng Qiu, Hao-Yi Zhang, Zong-Jun Yin, Da-Long Ren

**Affiliations:** College of Animal Science and Technology, Anhui Agricultural University, Hefei 230036, China; zr1212@stu.ahau.edu.cn (R.Z.); dingchenchen@stu.ahau.edu.cn (R.-C.D.); 21112204@stu.ahau.edu.cn (Q.Y.); 21720408@stu.ahau.edu.cn (C.-Z.Q.); astomble@stu.ahau.edu.cn (H.-Y.Z.)

**Keywords:** ROS, metformin, neutrophil recruitment, H3K18 lactylation, zebrafish

## Abstract

Beyond its well-established role in diabetes management, metformin has gained attention as a promising therapeutic for inflammation-related diseases, largely due to its antioxidant capabilities. However, the mechanistic underpinnings of this effect remain elusive. Using in vivo zebrafish models of inflammation, we explored the impact of metformin on neutrophil recruitment and the underlying mechanisms involved. Our data indicate that metformin reduces histone (H3K18) lactylation, leading to the decreased production of reactive oxygen species (ROS) and a muted neutrophil response to both caudal fin injury and otic vesicle inflammation. To investigate the precise mechanisms through which metformin modulates neutrophil migration via ROS and H3K18 lactylation, we meticulously established the correlation between metformin-induced suppression of H3K18 lactylation and ROS levels. Through supplementary experiments involving the restoration of lactate and ROS, our findings demonstrated that elevated levels of both lactate and ROS significantly promoted the inflammatory response in zebrafish. Collectively, our study illuminates previously unexplored avenues of metformin’s antioxidant and anti-inflammatory actions through the downregulation of H3K18 lactylation and ROS production, highlighting the crucial role of epigenetic regulation in inflammation and pointing to metformin’s potential in treating inflammation-associated conditions.

## 1. Introduction

Inflammation serves as the immune system’s innate response to injury or pathogenic threats, aiming to prevent the spread of disease and facilitate tissue repair [1]. Neutrophils, the front-line defenders in immune responses, play a critical role in eliminating external threats [2,3]. They not only engulf necrotic cells to inhibit further immune cell recruitment but also secrete mediators to spur growth and angiogenesis while producing lytic and protective enzymes to accelerate tissue repair [1,4,5]. Gaining insights into the regulatory mechanisms governing neutrophil-mediated inflammation could offer novel therapeutic avenues for diseases characterized by abnormal neutrophil activation.

Metformin (Met), a well-known oral antidiabetic medication, not only reduces insulin resistance and hepatic glucose production but also modulates immune responses [6,7]. It dampens cytokine production in macrophages and lymphocytes [8,9] and inhibits the NF-κB signaling pathway [10]. Moreover, metformin promotes the proliferation of regulatory T cells, which are crucial for tempering overactive immune responses [11,12]. Despite these known effects, the specific impact of metformin on neutrophilic inflammatory behavior has not been fully explored. Due to the optical transparency exhibited by zebrafish during their embryonic and larval stages, they serve as an ideal model for studying innate immunity. We aimed to leverage this advantage of zebrafish larvae by utilizing the transgenic zebrafish model *Tg(lyz:EGFP*). This model enables us to investigate whether metformin affects the recruitment of neutrophils to inflammatory sites in organisms.

Metformin’s capacity to modulate reactive oxygen species (ROS) has already been established; it curtails ROS production while bolstering antioxidant defenses through the inhibition of mitochondrial complex I and activation of AMP-activated protein kinase (AMPK) [13,14,15]. Excessive levels of ROS reportedly induce the abnormal aggregation of neutrophils at inflammatory sites, thereby exacerbating the inflammatory process [16,17]. To further investigate the relationships among metformin, inflammation, and ROS, we plan to conduct further experiments using an inflammation-induced zebrafish model. By monitoring the dynamic changes in ROS levels following metformin treatment, we aimed to elucidate the specific mechanisms through which metformin regulates ROS. This study provides a more comprehensive understanding of the role of metformin in neutrophil migration and the inflammatory process. These insights will contribute to unraveling the precise immunomodulatory mechanisms of metformin, helping to elucidate the treatment of relevant inflammatory diseases.

Metformin, a frontline medication for treating diabetes, may impact the body’s glycolytic metabolism and genes related to glycolysis via its mechanism of blood glucose control [18]. Lactic acid plays a crucial role in the glycolytic process, serving as a metabolic byproduct that significantly regulates histone lactylation levels [19]. Histone lactylation is a recently discovered translational modification associated with various biological processes, including tumor development, neural development, and inflammation [20]. We investigated whether metformin influences histone lactylation, thereby modulating neutrophil migration and affecting inflammation and immune responses in the body. In summary, our study employs zebrafish imaging to explore how metformin influences neutrophil recruitment. For the first time, we unveil the role of histone lactylation in regulating ROS production, thereby influencing the modulatory effect of metformin on neutrophil activity. Our findings introduce a novel mechanism for ROS regulation, enriching the current understanding in the field of ROS biology and expanding our insights into how metformin modulates immune responses.

## 2. Materials and Methods

### 2.1. Experimental Animals

Adult wild-type (WT/AB) and transgenic Tg(*lyz:EGFP*) strains were raised in an automatic circulating water system with a 14 h light and 10 h dark cycle at 28.5 °C. The system automatically monitors and adjusts the pH and salt ion concentration of the water, maintaining it at pH 7.0–7.5, and a salt ion concentration of approximately 500–550 mg/mL. Wild-type (WT) and transgenic Tg(*lyz:EGFP*) strains were naturally mated to obtain embryos. The zebrafish embryos were cultured in Hank’s solution at a constant temperature of 28.5 °C and pH 6.5–7.5 for 3–5 days. N-phenylthiourea (PTU; Sigma-Aldrich, St. Louis, MO, USA) was used to prevent pigment formation in the larvae [21]. We strictly followed the guidelines and regulations of Anhui Agricultural University Animal Resource Center SYXK (Anhui) 2016-007 [21].

### 2.2. Pharmacological Treatment

Metformin (Met) (Aladdin, Shanghai, China) was dissolved in ddH_2_O to prepare a stock solution (10 mM). To assess the toxicity of Met on zebrafish development, continuous soaking treatment was performed using different concentrations of metformin (0 μM, 50 μM, 100 μM, 400 μM, and 800 μM). After evaluating toxicity through the hatching rate, body length, and mobility of the zebrafish, Met (50 μM) was selected for subsequent experiments, while the cells were treated with a concentration of 10 μM for 8 h. To establish a systemic inflammation model, we injected LPS (0.15 mg/mL, 50 nL) (Sigma, L2630) into otic vesicles in zebrafish via microinjection. Cell inflammation was induced by exposure to LPS (2.5 μg/mL) for 2 h. Lactic acid (50 mM) (Macklin, China) was used to treat zebrafish at 5 dpf (days postfertilization) for 10 h, while the cells were treated with lactic acid (25 mM) for 6 h to increase the level of lactylation. H_2_O_2_ (200 μM) was used to treat zebrafish at 5 dpf for 2 h to increase the level of reactive oxygen species within the body, while the cells were treated with a concentration of 25 mM for 0.5 h.

### 2.3. Behavior Test

The short-term motor behavior of the zebrafish larvae (5 dpf) was evaluated using the Photomotor Response (PMR) model with a Viewpoint instrument (ViewPoint, Lyon, France). Briefly, the zebrafish larvae were placed in a 48-well plate for testing, with 10 larvae per group. The experimental procedure was as follows: 30 min of darkness, 5 min of light, and 5 min of darkness for three cycles [22]. The instrument was placed in an incubator at a constant temperature of 28.5 °C. An automatic video tracking system (ViewPoint, Lyon, France) was used to monitor larval movement for 30 min, and larval movement was recorded with Zebralab 3.11 software (ViewPoint, Lyon, France).

### 2.4. Morphological Observation

A tail swing rate (time/1 min) at 30 hpf, heart rate (time/20 s) at 48 hpf and 72 hpf, body length (cm) at 96 hpf, and survival rate at 96 hpf were observed under a stereomicroscope. Approximately 30 zebrafish were used for each experiment [23].

### 2.5. RNA Extraction and qRT-PCR

Total RNA was extracted from 35 zebrafish larvae and murine neutrophil cells using a SPARKeasy Tissue/Cell RNA Rapid Extraction Kit (SPARKjade, AC0202, Shandong, China), and reverse transcription was performed using a SPARKScript RT plus Kit (SPARKjade, Shandong, AG0304). Real-time quantitative PCR (Q-PCR) was conducted using a SYBR Green kit (RR820A, Takara, Japan). Approximately 40 cycles of 95 °C for 10 s and 60 °C for 30 s were used to amplify the related genes, including *tnf-α, il-1, il-6, cxcl8a, duox, sod1, cat, pkma, hdac3,* and *gapdh* (Appendix A). The Q-PCR experiments were repeated with three individual biological samples. The data were normalized to the expression level of the housekeeping gene β-actin, and the relative expression levels were calculated using the 2^(−ΔΔCt)^ method.

### 2.6. Caudal Fin Injury and Imaging

The Tg(*lyz:EGFP*) larvae (5 dpf) were anesthetized with a 0.1 g/mL MS-222 solution (Sigma, E10521). The tail fins were surgically damaged using blades under a stereomicroscope. Three hours after the injury, the migration of neutrophils in the caudal fin was observed using a fluorescence microscope, and the number of neutrophils was analyzed with ImageJ software (version 1.6.0) [21]. A total of 20 zebrafish were included in each group.

### 2.7. Otic Vesicle Inflammation Model

Zebrafish of the Tg(*lyz:EGFP*) strain were anesthetized with a 0.1 g/mL MS-222 solution (Sigma, E10521), and approximately 50 nL of LPS (0.2 μg/L, Sigma) was injected into the otic vesicle of the juvenile zebrafish. Three hours after LPS injection, the recruitment of neutrophils into the otic vesicle was observed using a fluorescence microscope, and the number of neutrophils was analyzed with ImageJ software (version 1.6.0). A total of 20 zebrafish larvae were used in each experimental group [22].

### 2.8. ROS Assay

DCFH-DA (Beyotime, China) was used to detect ROS production in both cells and zebrafish. For cell detection, NIH/3T3 cells were first inoculated into 12-well plates at a density of 2.5 × 10^4^ cells/well. Following drug treatment, the cells were incubated with approximately 500 μL of DCFH-DA (10 μM) at 37 °C for 30 min. After incubation, the cells were washed twice with 1x PBS to remove unpenetrated DCFH-DA. The fluorescence was then measured in FITC detection mode (488/525 nm) with a multifunctional microplate reader (VICTOR Nivo, PerkinElmer, Waltham, MA, USA), and images were captured using fluorescence microscopy (Carl Zeiss, Jena, Germany). For zebrafish, larvae were incubated with DCFH-DA (10 μM) at 28 °C for 25 min and then washed twice with PBS to remove unpenetrated DCFH-DA. Finally, images were captured under a fluorescence microscope, and the fluorescence intensity of the juvenile fish was quantified using ImageJ software (version 1.6.0) [24].

### 2.9. Lactate Content Assessment

The lactate content of the zebrafish larvae and cells was determined using a lactate assay kit (Abbkine, Wuhan, China). Briefly, after drug treatment, 30 larval samples were collected, and approximately 5 × 10^6^ cells were collected. The samples were added to the extraction mixture, mixed well, and then centrifuged at 12,000× *g* for 10 min. The supernatant was subsequently placed in a spectrophotometer at 565 nm to measure the absorbance [25].

### 2.10. Cell Culture

The murine macrophage cell line (RAW264.7) was purchased from Pricella (Wuhan, China). RAW264.7 cells were cultured in *α-MEM* (Hyclone, Logan, UT, USA) supplemented with 10% fetal bovine serum (FBS; Kang Yuan Biology, Tianjin, China) and then placed in a cell incubator (5% CO_2_, 37 °C). All the experiments were performed using 3–7 generations of cells.

### 2.11. Western Blotting

The lactation level of histone protein after treatment with different drugs was analyzed. After drug treatments, zebrafish larvae and cells from each group were collected, centrifuged, and lysed in RIPA buffer (Servicebio, Wuhan, China) for Western blotting. The collected samples were boiled for 5 min and electrophoresed on a 12.5% SDS-PAGE gel. The nitrocellulose membrane was blocked with 5% milk and incubated overnight with antibodies against H3K18la (1:1000, PTM-1406, PTMBIO), histone H3 (1:1000, PTM-7093, PTMBIO) or pan-Kla (1:1000, PTM-1401, PTMBIO). After washing, the sections were incubated with an HRP-labelled secondary antibody (Sangon, Shanghai, China) for 2 h at room temperature and imaged with chemiluminescent solution [26].

### 2.12. Statistical Analysis

The data are expressed as the mean ± SD. Comparisons between two groups were made with an unpaired *t* test, and comparisons between more than two groups were made with one-way ANOVA followed by the Bonferroni posttest. Survival rates are expressed as percentages, and the log-rank test was used to detect differences in zebrafish survival [27]. All analyses were performed with GraphPad software (GraphPad Software, Version 8.0.2). *p* < 0.05 was considered to indicate statistical significance (**** *p* < 0.0001, *** *p* < 0.001, ** *p*  <  0.01, * *p*  <  0.05).

## 3. Results

### 3.1. Metformin Reduces the Recruitment of Neutrophils to Sites of Inflammation

To determine the optimal concentration of metformin for treating zebrafish larvae, we immersed 24 hpf zebrafish larvae in solutions containing varying concentrations of metformin. By continuously monitoring the survival rate, heart rate, tail flick, body length, and activity over 96 hpf, we evaluated the impact of metformin on the growth and development of zebrafish larvae. Compared to the control, 50 μM Met had no significant impact on the survival rate (96 hpf), tail flick frequency (30 hpf), heart rate (24 hpf, 48 hpf, 72 hpf), or body length (96 hpf) of the zebrafish larvae (Figure 1b–h). Under an in vivo microscope, after metformin treatment, the overall morphology of the larvae did not significantly differ from that of the control group (Figure 1g). Additionally, we conducted a PMR experiment on zebrafish larvae using a behavioral detector. Compared to those in the control group, under both illuminated and dark conditions, 50 μM Met had no significant effect on the activity of the zebrafish larvae (Figure 1i–l). In summary, these findings indicate that 50 μM Met has no significant effect on the basic physiological functions or growth development of zebrafish larvae.

Subsequently, using transgenic zebrafish *Tg*(*lyz:EGFP*) marked with fluorescent neutrophils, we induced two acute inflammation models: tail fin injury and otic vesicle injection. We assessed, through in vivo imaging, whether metformin treatment affects neutrophil migration to inflamed sites. The results showed that in the tail fin injury model, metformin treatment significantly inhibited neutrophil migration to the injury site (Figure 2a,b). This was similarly validated in the otic vesicle model (Figure 2c,d). Proinflammatory cytokines are a group of proteins secreted by the body’s immune cells during an inflammatory response. *tnf-α, il-1β, il-6,* and *cxcl8a* are among these factors. They play a crucial role in the immune response and are commonly used to evaluate the level of inflammation in the body. After each larva was microinjected with LPS, they were treated with 50 μM metformin for 3 h, after which RNA was extracted to detect cytokine expression. The results demonstrated that, compared to that in the control group, the expression of proinflammatory cytokines in the LPS treatment group significantly increased (Figure 2e,h). However, compared with those in the LPS group, the expression in the groups pretreated with metformin was significantly lower. In the cellular experiments, we also detected the expression of the genes *tnf-α, il-1β, il-6,* and *cxcl8a* (Figure 2i,l). Consistent with the findings of previous studies, metformin reduced the levels of proinflammatory cytokines, suggesting its impact on intracellular immunity.

In summary, we demonstrated that metformin mitigates inflammatory reactions by downregulating neutrophil migration and reducing the levels of proinflammatory cytokines.

### 3.2. Metformin Reduces Reactive Oxygen Species Levels in Zebrafish and Cells

Studies suggest that the anti-inflammatory effects of metformin might be associated with its blockage of mitochondrial DNA synthesis and activation of the AMPK signaling pathway. We hypothesized that metformin might reduce the levels of reactive oxygen species (ROS) within the body. In this study, we explored the effects of metformin on ROS levels through both zebrafish experiments and cellular assays. First, in zebrafish experiments, we exposed zebrafish to hydrogen peroxide to induce the production of ROS. Then, another group of zebrafish was treated with metformin to assess its effect on ROS levels. The results showed that, in the metformin-treated group, ROS levels were significantly reduced (Figure 3a,b). We also examined the expression of key genes, such as *duox*, *sod1*, and cat, and found that metformin notably downregulated these genes (Figure 3c–f). These genes play a critical role in modulating ROS levels, suggesting that metformin effectively attenuates oxidative stress responses in organisms. Second, through cellular assays, we gauged the impact of metformin on ROS concentrations and observed that metformin notably decreased intracellular ROS levels. We investigated the expression profiles of key genes, including *duox, sod1*, and *cat*, which play pivotal roles in ROS modulation. After exposing cells to hydrogen peroxide, we administered metformin and monitored its influence on the expression of these genes. Remarkably, metformin treatment significantly inhibited the production of *duox*, an important enzyme that produces ROS, and downregulated the expression of antioxidant genes such as *cat* and *sod1*.

These results further support the protective role of metformin in reducing ROS-mediated cellular damage. By corroborating the results from these two perspectives, we concluded that metformin effectively reduces the ROS levels induced by hydrogen peroxide. These findings highlight the potential of metformin as a potential therapeutic drug for alleviating oxidative stress-related diseases and warrant further investigation into its mechanism of action in mitigating oxidative damage.

### 3.3. Reactive Oxygen Species Can Affect Neutrophil Migration and Immune Function

In our previous experiments, we measured the levels of reactive oxygen species inside zebrafish as well as the expression of the related genes *duox, sod1,* and *cat*. The findings revealed that treatment with hydrogen peroxide notably elevated the reactive oxygen species levels in zebrafish and resulted in the upregulation of *duox* gene expression. This finding suggests that hydrogen peroxide treatment may increase the production of reactive oxygen species by activating the Duox complex, leading to an increase in the intracellular oxidative stress response.

In further experiments, we deliberately increased the intracellular levels of reactive oxygen species to investigate their potential impact on neutrophil migration and immune function. We conducted experiments using two acute inflammation models, tail fin injury, and otolith vesicle injury to assess the effect of hydrogen peroxide treatment on neutrophil migration. Surprisingly, under conditions of elevated reactive oxygen species concentrations, neutrophil migration was significantly enhanced (Figure 4a–d). Moreover, hydrogen peroxide treatment induced the upregulation of the expression of these proinflammatory cytokines (*tnf-α, iL-1β, iL-6,* and *cxcl8a*) (Figure 4e–h), and the immune response was also strengthened. This finding is consistent with previous results showing elevated reactive oxygen species levels and upregulated expression of proinflammatory factors, suggesting that hydrogen peroxide might affect neutrophil migration by upregulating reactive oxygen species levels and the expression of proinflammatory factors. However, when we introduced metformin as an intervention, this effect was significantly suppressed. Although the levels of reactive oxygen species remained high, under the influence of metformin, both the migration capability of neutrophils and the immune response were significantly weakened compared to those in the untreated group. In the cell experiments, we also evaluated the effect of metformin on the levels of reactive oxygen species. Similarly, metformin significantly reduced the levels of proinflammatory factors (Figure 4i–l). These findings suggested that hydrogen peroxide treatment might induce the expression of proinflammatory factors by activating the oxidative stress response, leading to an inflammatory reaction, whereas metformin can inhibit this phenomenon.

### 3.4. Metformin Downregulates H3K18 Lactylation

In our study, we conducted an in-depth investigation into the functions of metformin, particularly its role in regulating cellular lactate production. In zebrafish experiments, we thoroughly examined the lactate content and the expression of genes such as *hdac3*, *pkma*, and *gapdh*.

The results showed that after metformin treatment, the lactate concentration significantly decreased (Figure 5a), indicating that metformin has a regulatory effect on lactate levels. Additionally, we observed changes in the expression levels of the hdac3, *pkma*, and *gapdh* genes, with their expression downregulated by metformin treatment (Figure 5b–d). These genes play crucial roles in the metabolism of lactate, and their downregulation implies a potential impact on the metabolic rate of lactate. These findings suggest that metformin may regulate the metabolism of lactate by affecting the expression of these lactate enzyme-related genes. At the protein level, we examined the lactation levels of zebrafish histone proteins and found that metformin significantly downregulated H3 histone lactation modifications, especially the H3K18 site (Figure 5e–g). In the cellular experiments, we also measured lactate levels and the expression of genes such as *hdac3, pkma*, and *gapdh*. Like in the zebrafish experiments, metformin treatment resulted in a significant decrease in the intracellular lactate concentration, suggesting its impact on cellular lactate metabolism (Figure 5h). Moreover, the expression of genes such as *hdac3, pkma*, and *gapdh* also changed (Figure 5i–k). These results support the view that metformin affects intracellular histone lactylation by regulating lactate enzyme-related genes. Combining the results from the zebrafish and cellular experiments, we concluded that metformin affects lactate enzyme-related genes in zebrafish/immune cells and downregulates histone lactylation.

These discoveries provide crucial insights into the mechanism by which metformin regulates metabolism.

### 3.5. H3K18 Lactylation Affects Neutrophil Recruitment by Increasing the Levels of Reactive Oxygen Species

In our previous research, we found that metformin had a significant effect on decreasing histone lactylation. Based on this discovery, we further explored the effect of lactate on lactylation and H3K18 lactylation, examining the role of metformin in this context. Western blot results clearly showed that, compared to those in the Con group, the lactate group exhibited significant increases in overall lactylation and H3K18 lactylation levels (Figure 6a–c). However, when lactate and metformin were combined, this lactate-induced upregulation was effectively inhibited. Additionally, lactate significantly enhanced the expression of the *pkma* and *gapdh* genes (Figure 6d,e). However, in the group treated with both lactate and metformin, the expression of these genes was notably suppressed, further demonstrating the ability of metformin to regulate lactylation modifications. Regarding reactive oxygen species, we noted that lactate not only increased the production of reactive oxygen species in zebrafish (Figure 6f,g) but also significantly upregulated the expression of the oxidative stress-related genes *duox* and *sod1*. These findings suggested that lactate might regulate the generation of reactive oxygen species by activating the oxidative stress pathway. Finally, we conducted experiments using two acute inflammation models, tail fin injury and otolith sac injury, to investigate the effects of lactate treatment on neutrophil migration. We observed that after lactate treatment in zebrafish, whether in tail cut or otolith sac injury experiments, the lactate group demonstrated a significant increase in neutrophils. The addition of metformin weakened neutrophil migration, consequently reducing the inflammatory response in the body (Figure 6j–q). These findings align with previous findings of increased reactive oxygen species levels, suggesting that lactate might influence neutrophil migration behavior by upregulating reactive oxygen species levels.

Overall, our research findings indicate that lactate affects neutrophil migration by elevating the levels of reactive oxygen species. Metformin can alleviate oxidative stress and inflammation induced by increased lactate. These findings provide crucial insights into the role of lactate in immune regulation and inflammation. Nonetheless, further investigations are warranted to clarify the regulatory interplay between lactate and reactive oxygen species and to unearth its potential application in treating inflammation-associated ailments.

In summary, our research emphasizes that hydrogen peroxide affects neutrophil migration by upregulating the levels of reactive oxygen species and might induce an inflammatory response during this process. This provides important clues for a deeper understanding of the role of oxidative stress in immune regulation and the inflammatory process. These findings clearly indicate that while reactive oxygen species can regulate the migration and immune function of neutrophils, metformin has potential regulatory effects that can effectively mitigate the biological effects of excessive production of reactive oxygen species. This finding offers new insights into the potential application of metformin to cellular immune regulation. However, further research is still needed to elucidate the regulatory mechanism between metformin and reactive oxygen species and to explore its potential therapeutic value in inflammation-related diseases. Our research provides valuable information in this field, laying a foundation for future studies in related areas.

## 4. Discussion

Our research provides a more nuanced understanding of metformin, conventionally celebrated for its antidiabetic capabilities, and suggests its potential as a versatile therapeutic tool. Specifically, we discovered that metformin modulates inflammation via the H3K18 lactylation/ROS axis, opening an exciting new line of inquiry. This epigenetic control mechanism offers promising avenues for novel therapies targeting diseases typified by unchecked ROS and inflammation.

Although the anti-inflammatory and antioxidant effects of metformin have been extensively demonstrated, the underlying molecular mechanisms have remained complex and challenging. Recent research has identified PEN2 as a direct molecular target of metformin, elucidating the molecular interaction-mediated lysosomal pathway and the specific activation of AMPK. This clarification underscores the crucial role of AMPK-related pathways in the diverse functions of metformin, including glucose reduction, lifespan extension, and anti-aging effects [28]. Moreover, the anti-inflammatory effects of metformin are also gradually gaining attention. Research has shown that metformin can significantly reduce the levels of *IL-1β* and *IL-6* but not inhibit the activation of the inflammatory pathway NF-κB [29]. In terms of immune regulation, metformin has been demonstrated to potentially modulate immune activity in the human body, including the inhibition of T-cell-mediated immune responses [30]. Additionally, metformin promoted innate immunity in mice through the conserved PMK-1/p38 MAPK pathway [31]. Our research findings align with these observations, as we have also substantiated the reduction in proinflammatory factors (TNF-α, IL-1β, IL-6, and IL-8) in zebrafish treated with metformin. These findings are consistent with prior studies on the anti-inflammatory effects of metformin (Jing et al., 2018; Xian et al., 2021) [32,33], further confirming the role of metformin in suppressing inflammatory responses.

The diverse mechanisms of action of metformin in the regulation of ROS have attracted widespread attention. Studies indicate that metformin can modulate ROS through various pathways. Notably, research reports highlight the ability of metformin to alleviate the accumulation of mitochondrial ROS (mtROS) in macrophages following NLRP3 activation [34]. Consequently, metformin has been confirmed to reduce ROS production by activating the AMPK pathway or directly affecting mitochondria [35]. In line with these findings, our research results align with those of Sekar et al., who demonstrated a significant reduction in high ROS production induced by hydrogen peroxide in both zebrafish and macrophages [36,37]. This discovery further solidifies the understanding of the effectiveness of metformin in regulating ROS levels, providing crucial clues for revealing additional details about its ability to regulate cellular activity.

ROS play a pivotal role in the modulation of neutrophil function [38]. Early studies suggest that an increase in ROS levels triggers a cascade of cellular responses, including the migration of neutrophils [17,39]. Moreover, elevated oxidative stress may induce changes in the inflammatory microenvironment and disrupt the tissue acid-base balance [40]. Notably, our research revealed that the addition of metformin leads to a reduction in lactate levels when the body is in an inflammatory state. Studies suggest that high concentrations of lactate may drive immune cells toward a more immunosuppressive state, influencing the regulation of inflammation [41,42]. Lactate may also impact the expression levels of inflammatory mediators by modulating cell signaling pathways such as the NF-κB pathway [42]. With increasing lactate concentrations, the level of lactylation in the body also increases. Previous research has suggested the association of H3K18 lactylation with certain cellular stress responses [19,43]. However, our study clearly demonstrated the relationship between lactate and reactive oxygen species levels. Our data further affirm the central role of ROS in inflammation and link ROS with H3K18 lactylation, unveiling a complex feedback loop in which subtle molecular changes profoundly impact cellular behavior and overall health. This discovery not only expands our understanding of the interaction between ROS and metabolic products but also provides a new perspective for a deeper comprehension of the molecular mechanisms governing inflammation regulation.

Interestingly, our study revealed that metformin can reduce the expression of histone H3K18 and lactylation genes, indicating that metformin can reduce the level of lactylation. This is a new discovery, and no previous literature has reported that metformin can affect lactylation, which opens up new avenues for in-depth study of the molecular mechanism of metformin’s anti-inflammatory effect. More importantly, our study reveals a potential link between lactylation and the generation of reactive oxygen species. We observed that the addition of lactic acid could augment ROS levels, subsequently affecting neutrophil migration, and that this effect could be mitigated by metformin. This finding suggested that lactylation might affect neutrophil function by influencing ROS production and that metformin may play an anti-inflammatory role by modulating this process. Finally, our work showed that hydrogen peroxide treatment elevated proinflammatory factor expression in zebrafish, amplifying neutrophil migration and underscoring the critical role of ROS in inflammation.

In conclusion, as we continue to untangle the complex interplay between immunity, inflammation, and metabolism, drugs such as metformin, which influence these pathways, may serve as cornerstones for new therapeutic approaches. Future work should explore the precise mechanisms and potential applications of H3K18 lactylation, thereby enriching this burgeoning field of study.

## 5. Conclusions

In summary, metformin effectively reduces neutrophil recruitment in zebrafish inflammation models by inhibiting histone (H3K18) lactylation, leading to a subsequent decrease in reactive oxygen species (ROS) levels. The direct linkage between histone lactylation and ROS generation unveils promising anti-inflammatory targets, emphasizing the potential therapeutic implications of metformin in immune-related disorders.

## Figures and Tables

**Figure 1 antioxidants-13-00176-f001:**
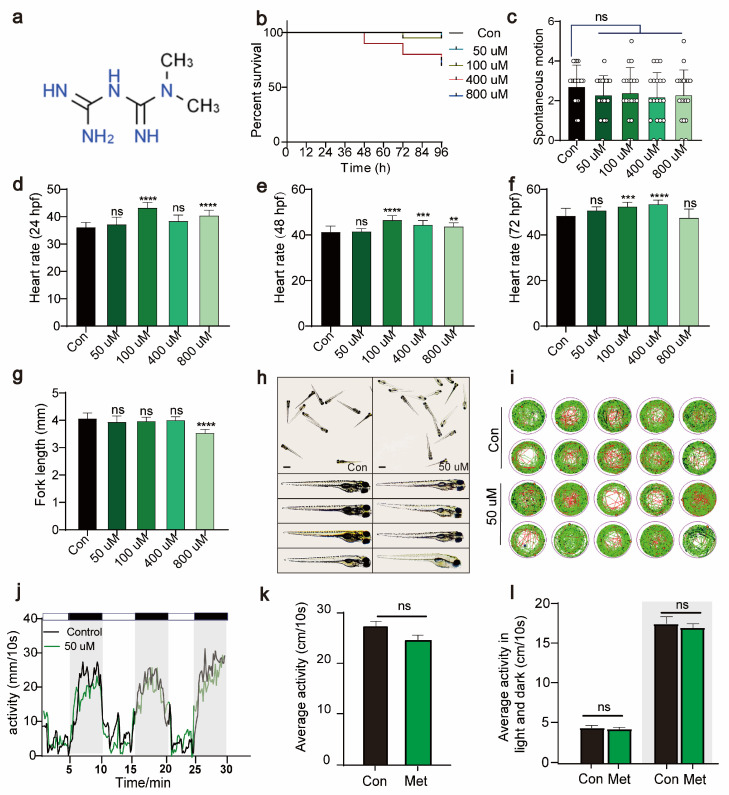
Metformin has no significant effect on the growth, development, or activity of zebrafish larvae. (**a**) The chemical formula of metformin. (**b**) Survival rate curve of zebrafish larvae from 0 to 96 hpf (%survival, n = 15). (**c**) Tail swing frequency of 30 hpf zebrafish larvae (swing per minute, n = 15). (**d**) Heartbeat of 24 hpf zebrafish larvae (beats per minute, n = 30). (**e**) Heartbeat of 48 hpf zebrafish larvae (beats per minute, n = 30). (**f**) Heartbeat of 24 hpf zebrafish larvae (beats per minute, n = 30). (**g**) Statistical graph of the fork length of 96 hpf zebrafish larvae (mm, n = 15). (**h**) Morphological images of 96 hpf zebrafish larvae observed under a stereomicroscope (scale bar: 350 μm). There were no significant differences in morphology between the 50 μM metformin-treated group and the control group. The data were analyzed using one-way ANOVA and the log-rank test. All the experiments were repeated three times. (**i**,**j**) The behavioral rhythms of the control group (n = 10/group) and the 50 μM metformin group (n = 10/group) after 3 days were observed under conditions of 14 h of light and 10 h of alternating darkness (LD). (**k**) The behavioral rhythms of the Con group (n = 10/group) and Met group (n = 10/group) were observed on day 3 under continuous light (LL) conditions. (**l**) The 30 min behavioral rhythms of the Con group (n = 10/group) and Met group (n = 10/group) were observed under continuous light (LL) and continuous darkness (LD) conditions. (**** *p*  <  0.0001, *** *p*  <  0.001, ** *p*  <  0.01, “ns”, no statistical difference).

**Figure 2 antioxidants-13-00176-f002:**
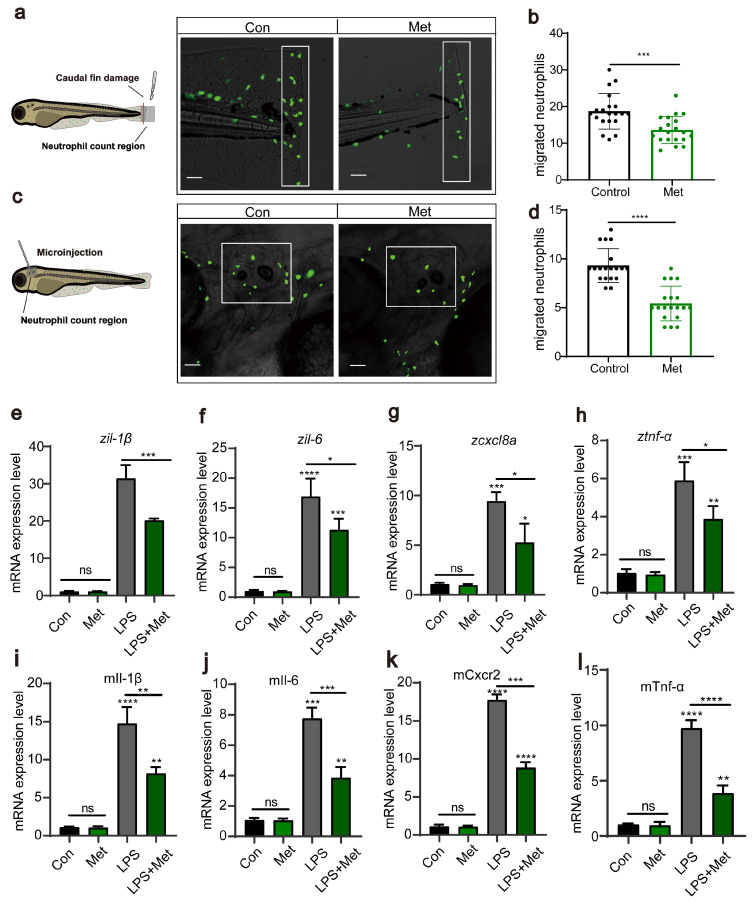
Metformin reduces the recruitment of neutrophils to inflammatory sites. (**a**,**c**) Pattern diagrams showing the caudal fin damage model and otic vesicle inflammation model in zebrafish infected with the Tg(*lyz:EGFP*) strain. Fluorescence images showing migrating neutrophils. The white rectangles indicate the counting area (scale bar: 200 μm). (**b**,**d**) Statistical analysis showed that the number of neutrophils recruited to the injury site was significantly lower in the larvae-fed metformin (n = 20). (**e**–**h**) Q-PCR analysis showed that the expression levels of the inflammation-related genes *il-1β, il-6, cxcl8a,* and *tnf-α* were significantly downregulated after metformin treatment in juvenile fish. (**i**–**l**) Q-PCR analysis showed that the expression levels of the inflammation-related genes *il-1β, il-6, cxcl8a,* and *tnf-α* in murine macrophages under metformin conditions were significantly downregulated. (**** *p* < 0.0001, *** *p* < 0.001, ** *p*  <  0.01, * *p*  <  0.05, “ns”, no statistical difference).

**Figure 3 antioxidants-13-00176-f003:**
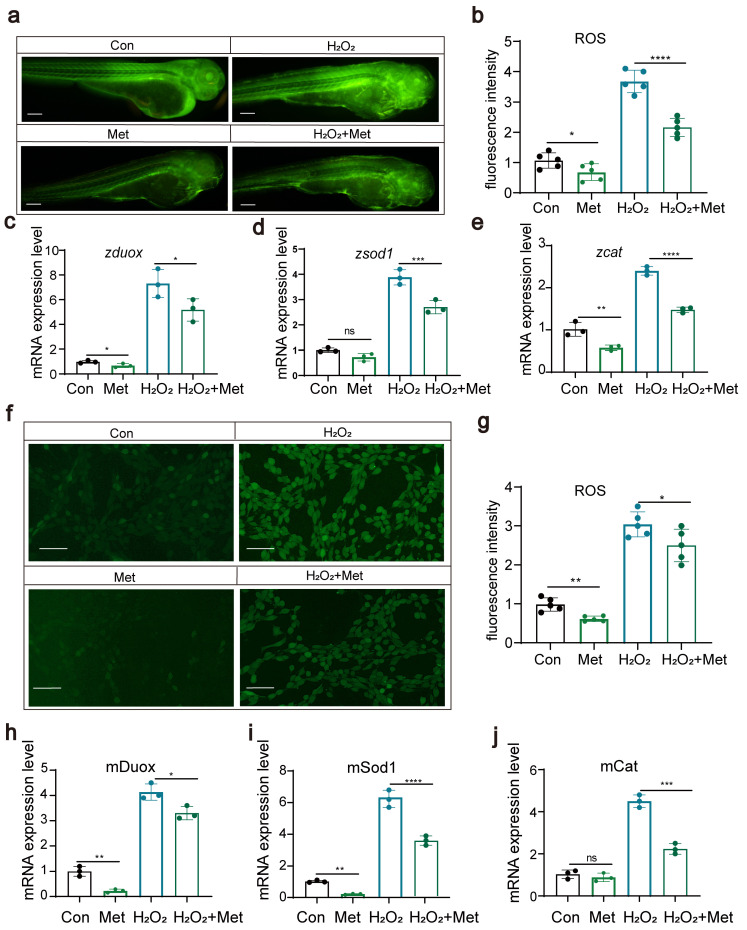
Metformin can reduce reactive oxygen species levels in zebrafish and cells (**a**,**b**). The results of fluorescence staining showed that metformin downregulated the level of ROS in zebrafish and that H_2_O_2_ upregulated the level of ROS (scale bar: 200 μm). The addition of metformin alleviated the effect of H_2_O_2_ on reactive oxygen species in zebrafish (n = 25). (**c**–**e**) Q-PCR results showing that metformin treatment reduced the upregulation of H_2_O_2_-induced expression of *duox*, sod1, and cat in zebrafish. (**f**,**g**) Fluorescence staining results showing that metformin downregulated ROS and that H_2_O_2_ upregulated ROS in murine macrophages. The addition of metformin alleviated the effect of H_2_O_2_ on reactive oxygen species in murine macrophages. (**h**–**j**) Q-PCR results showing that metformin treatment reduced the upregulation of H_2_O_2_-induced expression of duox, sod1, and cat in murine macrophages. (**** *p*  <  0.0001, *** *p*  <  0.001, ** *p*  <  0.01, * *p*  <  0.05, “ns”, no statistical difference).

**Figure 4 antioxidants-13-00176-f004:**
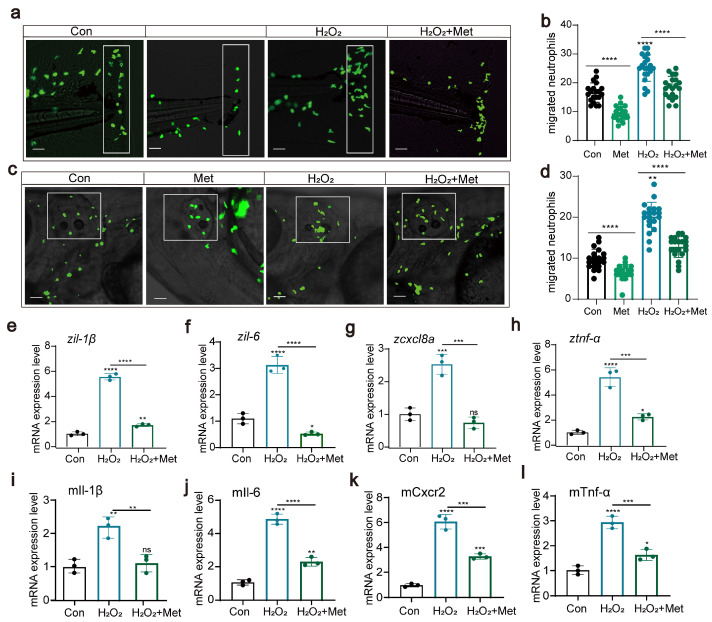
Reactive oxygen species can affect neutrophil migration and immune function (**a**–**d**) Pattern diagrams showing the caudal fin damage model and otic vesicle inflammation model using zebrafish from the Tg(*lyz:EGFP*) strain. Fluorescence images showing migrating neutrophils. The white rectangles indicate the counting area (scale bar: 200 μm). H_2_O_2_ treatment significantly increased neutrophil recruitment to the injury site, and metformin mitigated this effect (n = 20). (**e**–**h**) Q-PCR results showing that metformin treatment decreased the upregulation of the inflammatory genes *il-1β, il-6, cxcl8a,* and *tnf-α* induced by H_2_O_2_ in juvenile zebrafish. (**i**–**l**) Q-PCR results showed that metformin treatment decreased the upregulation of the inflammatory genes *il-1β, il-6, cxcl8a,* and *tnf-α* induced by H_2_O_2_ in murine macrophages. (**** *p*  <  0.0001, *** *p*  <  0.001, ** *p*  <  0.01, * *p*  <  0.05, “ns”, no statistical difference).

**Figure 5 antioxidants-13-00176-f005:**
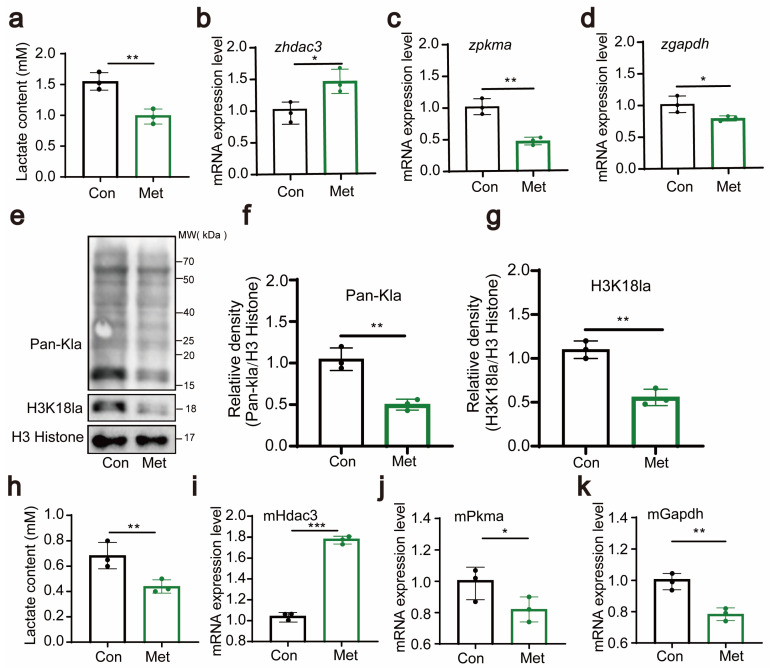
Metformin downregulates H3K18 lactylation. (**a**) Lactate levels in zebrafish larvae significantly decreased after treatment with metformin. (**b**–**d**) Q-PCR analysis showed that the lactate metabolism-related gene hdac3 was significantly upregulated and that the expression levels of *pkma* and *gapdh* were significantly downregulated in zebrafish larvae after metformin treatment. (**e**–**g**) Western blotting results showed that the levels of histone lactylation and histone lactylation (H3K18) were significantly reduced in zebrafish larvae after metformin treatment. (**h**) Lactate levels in murine macrophages significantly decreased after treatment with metformin. (**i**–**k**) Q-PCR analysis showed that the lactate metabolism-related gene hdac3 was significantly upregulated and that the expression levels of *pkma* and *gapdh* were significantly reduced in murine macrophages after metformin treatment. (*** *p*  <  0.001, ** *p*  <  0.01, * *p*  <  0.05).

**Figure 6 antioxidants-13-00176-f006:**
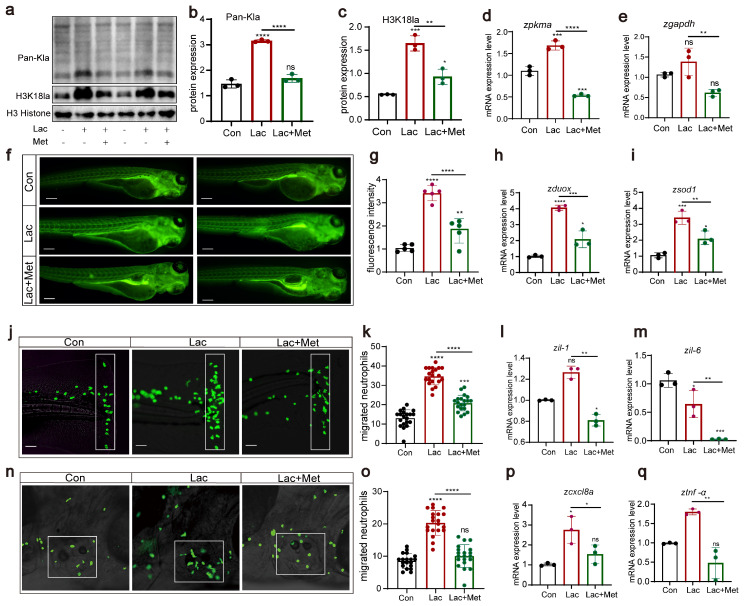
H3K18 lactylation affects neutrophil recruitment by increasing the levels of reactive oxygen species (**a**–**c**) Western blotting results showed that the levels of histone lactylation and histone lactation (H3K18) in zebrafish larvae after lactic acid treatment were significantly increased, and this lactate-induced upregulation was effectively inhibited when lactic acid was combined with metformin. (**d**,**e**) Q-PCR results showed that the levels of the *pkma* and *gapdh* genes related to lactic acid metabolism were significantly increased after lactic acid treatment, and this lactate-induced upregulation was effectively inhibited when lactic acid was combined with metformin. (**f**,**g**) Fluorescence staining results showing that lactic acid upregulated ROS levels in zebrafish. The addition of metformin alleviated the effect of hydrogen peroxide on reactive oxygen species in zebrafish (n = 25). (**h**,**i**) Q-PCR results showing that metformin treatment decreased the upregulation of *dox* and *sod1* in lactate-induced zebrafish. (**j**,**k**,**n**,**o**) Pattern diagrams showing the caudal fin damage model and otic vesicle inflammation model generated from zebrafish infected with the Tg(*lyz:EGFP*) strain. Fluorescence images showing migrating neutrophils. The white rectangles indicate the counting area (scale bar: 200 μm). Lactic acid treatment significantly increased neutrophil recruitment to the injury site, and metformin mitigated this effect (n = 20). (**l**,**m**,**p**,**q**) Q-PCR results showing that metformin treatment decreased the upregulation of the inflammatory genes *il-1β, il-6, cxcl8a,* and *tnf-α* induced by lactic acid in juvenile zebrafish. (**** *p*  <  0.0001, *** *p*  <  0.001, ** *p*  <  0.01, * *p*  <  0.05, “ns”, no statistical difference).

## Data Availability

The data are contained within the article and Appendix A.

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
