# Peer review of "Metformin Attenuates Neutrophil Recruitment through the H3K18 Lactylation/Reactive Oxygen Species Pathway in Zebrafish"

_antioxidants, 2024, doi:10.3390/antiox13020176_

Round 1

Reviewer 1 Report

Comments and Suggestions for Authors

The manuscript by Zhou et al, "metformin attenuates neutrophil recruitment through HsK18 lactylation/ROS pathway in zebrafish", presents a set of high-quality results, using methodologies that are appropriate for the objectives in question. The results are clearly presented. However, the following improvements should be considered:

Minor points:

i) Figure 1 could be presented as a supplementary figure;

ii) it seems more logical to place figure 6 after figure 3. In figure 3 it shows that metformin reduces the levels of reactive oxygen species and then (in what is now figure 6) it would show that reactive oxygen species influence neutrophil migration, then showing that metformin inhibits H3K18 lactylation (figure 4) and that H4K18 lactylation affects neutrophil recruitment by increasing the formation of reactive oxygen species. 

In my opinion, this alignment of the figures seems easier to follow and more in line with the study's main message: metformin inhibits neutrophil recruitment by inhibiting H3K18 lactylation.

iii) The authors claim that metformin inhibits the expression of H3K18 (line 363 - 364). Where can this effect be observed? According to what is shown in figure 5, the effect of metformin alone has not been studied on HsK18 expression.

Major points:

What determines the action of a drug is its concentration. The authors investigated the effect of metformin, which is used clinically as an oral antidiabetic. They studied a single concentration (50µM), which was the lowest concentration and showed no influence on the parameters studied and shown in figure 1. However, 50 µM of metformin is a higher concentration than has been reported for plasma levels in humans when metformin is used as an antidiabetic.  This point should be taken into account when the authors try to discuss the relevance of the results to support the clinical usefulness of this drug at this concentration.

The introduction should be thoroughly modified. The introduction is supposed to present the working hypothesis based on the current state of knowledge. However, the authors have already introduced data from the present study into this presentation, creating confusion for the reader as to whether we are reading the introduction or the discussion of the results. In the present format, the reader can hardly understand what is the hypothesis under study.

The discussion is still very preliminary. It needs to be improved, thoroughly reorganised by actually discussing each of the study's main results and integrating this knowledge into the current state of the art.

The authors should also take into account the various effects of metformin, particularly on the immune system and on lactate formation, studies with metformin in zebrafish and the role of lactylation in post-translational or epigenetic changes. As it is, the reader will hardly see this section as an objective discussion of the results presented.

Reviewer 2 Report

Comments and Suggestions for Authors
  1. The title should not contain any short form.
  2. The version of all software needs to be mentioned.
  3. The abstract needs to be modified in a more logical way.
  4. "Significantly reduces": How do the authors define this?
  5. Epigenetic regulation: What is the relationship with this study?
  6. The introduction is poor. No problem statement is found; it needs to be changed entirely.
  7. Metformin's capacity to modulate reactive oxygen species: What is the significance of this paragraph?
  8. Salt ion concentration of approximately 500-550 mg/ml: How is this determined?
  9. "We strictly followed the guidelines and regulations of Anhui Agricultural University Animal Resource Center SYXK (Anhui) 2016-007": A reference is needed.
  10. Few references are used in the methods.
  11. Results and conclusion: The section devoted to the explanation of the results suffers from the same problems revealed so far. Your storyline in the results section (and conclusion) is hard to follow. Moreover, the conclusions reached are far from what one can infer from the empirical results.
  12. The discussion should be organized around arguments, avoiding simply describing details without providing much meaning. A real discussion should also link the findings of the study to theory and/or literature.
  13. Authors need to improve their writing style. Additionally, the whole manuscript needs to be checked by native English speakers.
  14. The conclusion does not contain any information.
  15. Figure 4: The Western result is confusing.
  16. Figure 7: The schematic of the mechanism by which metformin reduces neutrophil recruitment does not make any sense.
Comments on the Quality of English Language

Authors need to improve their writing style. Additionally, the whole manuscript needs to be checked by native English speakers.

Round 2

Reviewer 1 Report

Comments and Suggestions for Authors

The authors addressed the main issues raised in the first version of the manuscript and modified the text accordingly. The introduction and discussion were improved. In my opinion the authors could have gone further in this improvement, but, in the present version the introduction is already in a form that allows the objective of the study of the scientific problem that they intend to investigate, and the discussion an interpretation of the results in relation to the most recent works that have investigated the mechanism of action of metformin. Above all, the value of the work lies in demonstrating the modulation of lactylation by metformin and the importance that this mechanism may have in understanding its effects. 

Author Response

RESPONSE: Thank you for your feedback. We appreciate your valuable insights and suggestions for improving our manuscript, titled "Metformin attenuates neutrophil recruitment through the H3K18 lactylation/reactive oxygen species pathway in zebrafish" (Manuscript ID: 2800762). We are pleased to learn that the revisions made to the introduction and discussion sections have contributed positively to the overall clarity of the manuscript. We acknowledge the reviewer's comments regarding the potential for further improvement in these sections. I have thoroughly revised the manuscript in accordance with the suggestions and feedback. To ensure transparency, all modifications have been highlighted in red for your convenience. These changes include improvements to the introduction and discussion sections, as well as a more in-depth exploration of the main topics raised by the reviewers. I appreciate the valuable insights provided by the reviewers and the editorial team, which have undoubtedly contributed to the enhancement of our manuscript. I am confident that the revisions have strengthened the overall quality and clarity of the content. I would like to express my gratitude for your continued support and guidance throughout this process. I look forward to the opportunity for the revised manuscript to undergo further evaluation. Thank you for your time and consideration.

Reviewer 2 Report

Comments and Suggestions for Authors

I am not satisfied with the method section. Need to add more relevant citations. Introduction introduction and discussion section look good to me. 

Comments on the Quality of English Language

Minor revisions

Author Response

RESPONSE:I would like to express my sincere gratitude for the time and effort you have dedicated to evaluating my manuscript, titled "Metformin attenuates neutrophil recruitment through the H3K18 lactylation/reactive oxygen species pathway in zebrafish." I have carefully considered theyour comments, particularly regarding the method section. In response to your suggestion, I have made extensive revisions to the method section, incorporating additional relevant citations to enhance its comprehensiveness and clarity. I am pleased to note that you found the introduction and discussion sections satisfactory, and I appreciate their positive feedback in this regard. I believe the revisions have strengthened the overall quality of the manuscript. I am committed to ensuring that the research is presented with the highest standards of rigor and scholarship. I would like to express my gratitude once again for the constructive feedback, which has undoubtedly contributed to the improvement of the manuscript. I am confident that these enhancements will enhance the clarity and scholarly value of the paper. Thank you for your continued support throughout this review process. I look forward to any further guidance you may provide.